# Serum Chemerin Does Not Differentiate Colorectal Liver Metastases from Hepatocellular Carcinoma

**DOI:** 10.3390/ijms20163919

**Published:** 2019-08-12

**Authors:** Susanne Feder, Arne Kandulski, Doris Schacherer, Thomas S. Weiss, Christa Buechler

**Affiliations:** 1Department of Internal Medicine I, Regensburg University Hospital, 93053 Regensburg, Germany; 2Children’s University Hospital (KUNO), Regensburg University Hospital, 93053 Regensburg, Germany

**Keywords:** alpha-fetoprotein, liver steatosis, hypertension

## Abstract

The chemoattractant adipokine chemerin is related to the metabolic syndrome, which is a risk factor for different cancers. Recent studies provide evidence that chemerin is an important molecule in colorectal cancer (CRC) and hepatocellular carcinoma (HCC). Serum chemerin is high in CRC patients and low in HCC patients and may serve as a differential diagnostic marker for HCC and liver metastases from CRC. To this end, serum chemerin was measured in 36 patients with CRC metastases, 32 patients with HCC and 49 non-tumor patients by ELISA. Chemerin serum protein levels were, however, similar in the three cohorts. Serum chemerin was higher in hypertensive than normotensive tumor patients but not controls. Cancer patients with hypercholesterolemia or hyperuricemia also had increased serum chemerin. When patients with these comorbidities were excluded from the calculation, chemerin was higher in CRC than HCC patients but did not differ from controls. Chemerin did not correlate with the tumor markers carcinoembryonic antigen, carbohydrate antigen 19-9 and alpha-fetoprotein in both cohorts and was not changed with tumor-node-metastasis stage in HCC. Chemerin was not associated with hepatic fat, liver inflammation and fibrosis. To conclude, systemic chemerin did not discriminate between CRC metastases and HCC. Comorbidities among tumor patients were linked with elevated systemic chemerin.

## 1. Introduction

Colorectal cancer (CRC) is the third most prevalent cancer worldwide and a leading cause of tumor-related mortality. The liver is a common site for CRC metastasis [1,2]. Hepatocellular carcinoma (HCC) typically develops in the cirrhotic liver, but about 20% arise in the non-cirrhotic liver [3]. Discrimination between secondary hepatocarcinoma and HCC may be challenging in those patients. Clinically it is, however, highly relevant to distinguish primary and metastatic liver tumors. First, there are different therapies for patients with CRC metastases and HCC. Second, it is important to identify the primary tumor in metastatic disease [4], therefore biomarkers may be helpful in early diagnosis. Carcinoembryonic antigen (CEA) is already clinically used as diagnostic and prognostic marker in CRC [5], and systemic levels were indeed higher in secondary than primary liver tumors. Sensitivity of serum CEA for CRC metastases was 88% and 25% for HCC [1]. Cancer antigen 19-9 (CA19-9) had a 16% sensitivity for colon cancer and a 7.7% sensitivity for HCC and could be used as an additional prognostic tool [2,6]. Alpha-fetoprotein (AFP) is a diagnostic biomarker for HCC with a low sensitivity and specificity, and thus cannot differentiate between HCC and CRC metastases [1].

Recent studies described a role of chemerin in CRC pathophysiology and diagnosis [7,8,9]. Chemerin is a chemoattractant protein most abundant in adipocytes and hepatocytes [10]. Chemerin is released from the cells as a biological inert molecule, which is activated by C-terminal proteolysis. Chemerin attracts immune cells such as macrophages and natural killer cells [11]. Moreover, chemerin regulates adipogenesis, angiogenesis and glucose metabolism [12]. Chemerin expression was reduced in a variety of cancers, and was also low in colon adenomas [13].

High plasma chemerin predicted a greater risk of CRC. Notably, this association was still significant when CRC risk factors such as age, body mass index and dietary habits were considered [8]. A second study detected higher chemerin in patients with CRC compared to healthy controls. Here, serum chemerin positively correlated with tumor-node-metastasis (TNM) stage [7]. In colon cancer patients, chemerin was increased though it was not associated with TNM classification [9]. Sytemic chemerin was further positively related to the number of adenomas in patients with colorectal adenomas [14].

Chemerin also plays a role in hepatocellular carcinoma (HCC) and low expression in the tumor was an independent prognostic factor [15]. Similarly, circulating chemerin levels were about 20-fold reduced in HCC patients [16]. Chemerin was not related to HCC prognosis [17]. Negative correlations of chemerin with Child–Pugh score, alanine aminotransferase and bilirubin demonstrated a close and negative association of serum chemerin with hepatic function in patients with liver cirrhosis [17,18]. In contrast, chronic hepatitis C patients had higher serum chemerin compared to controls, which was surprisingly negatively correlated with biopsy proven necro-inflammation [19]. Likewise, chemerin was high in men with alcohol abuse [20]. In patients with non-alcoholic steatohepatitis (NASH) serum chemerin was either induced or normal [21]. Decline of serum chemerin thus happens particularly in patients with severely impaired liver function and possibly HCC.

Obesity, hyperglycemia, dyslipidemia and hypertension are components of the metabolic syndrome, and all of them were linked with the development of cancers [22]. Patients with non-alcoholic fatty liver disease (NAFLD) have a higher risk for gastrointestinal tumors and the underlying factor is most likely the close relationship between NAFLD and traits of the metabolic syndrome [23].

Of note, circulating chemerin was positively associated with all of the components of the metabolic syndrome [10,24,25]. Therefore, chemerin´s association with CRC may in part stem from the relationship between CRC and features of the metabolic syndrome [7,8,10,14,24,25]. Metabolic diseases also contribute to HCC development [22]. Whether chemerin correlates with traits of the metabolic syndrome in patients with cancers is, however, not well studied.

The liver is a common site of metastases from tumors arising in the gastrointestinal tract [26]. Here, we suggested that chemerin in serum may be appropriate to discriminate between colorectal liver metastases and HCC. A further aim was to identify associations of chemerin levels with components of the metabolic syndrome in patients with cancers.

## 2. Results

### 2.1. Association of Chemerin with Gender, Age and BMI

Serum chemerin was measured in 32 HCC, 36 CRC patients and 49 controls by ELISA (Table 1). Controls were patients which came to the hospital because of mostly epigastric or stomach pain but without any cancers [27,28]. HCC patients had higher bilirubin and aminotransferase activities than CRC patients in accordance with previous studies [29]. Levels of γ-glutamyltransferase were also increased in HCC patients (Table 1). Control cohort had lower aminotransferase activities than the group of HCC patients (Table 1). There were fewer female patients in the HCC compared to the control group (Table 1).

Chemerin was comparable in male and female tumor patients in the whole cohort (*p* = 0.354) and in the individual subgroups (*p* = 0.976 for controls, *p* = 0.540 for CRC and *p* = 0.511 for HCC). Levels neither correlated with age (*r* = 0.20, *p* = 0.10) nor BMI (*r* = −0.27, *p* = 0.83) in the tumor patients. This was also the case when the two cohorts of cancer patients were analyzed separately (CRC: age: *r* = 0.06, *p* = 0.73; BMI *r* = 0.09, *p* = 0.61; HCC: age: *r* = 0.30, *p* = 0.10; BMI *r* = −0.49, *p* = 0.79). In the controls chemerin positively correlated with age (*r* = 0.368, *p* = 0.009) but not with BMI (*r* = 0.059, *p* = 0.688).

### 2.2. Chemerin, CEA and CA19-9 in HCC and CRC Patients

Chemerin levels were similar in controls, HCC and CRC patients (Figure 1A). The tumor marker alpha-fetoprotein (AFP; known from 25 HCC patients and 15 CRC patients) did not differ between the two cohorts of tumor patients (*p* = 0.07). CEA (known from 20 HCC patients and 32 CRC patients) and CA19-9 (known from 19 HCC patients and 31 CRC patients) were higher in CRC patients (Figure 1B,C).

Chemerin did not correlate with AFP (*r* = −0.20, *p* = 0.22), CEA (*r* = 0.08, *p* = 0.57) and CA19-9 (*r* = 0.15, *p* = 0.31) in the cancer patients of the whole cohort and when both groups were analyzed separately (CRC: AFP *r* = −0.08, *p* = 0.77, CEA *r* = 0.18, *p* = 0.34 and CA19-9 *r* = 0.31, *p* = 0.09; HCC: AFP *r* = −0.15 *p* = 0.48, CEA *r* = −0.19, *p* = 0.41 and CA19-9 *r* = −0.42, *p* = 0.07 ). In the HCC group chemerin was not associated with tumor size (*r* = 0.271, *p* = 0.13), grade (*r* = 0.044, *p* = 0.82) or tumor-node-metastasis (TNM) stage (Figure 1D). Patients without vascular invasion had serum chemerin similar to those with this development (Figure 1E).

For the CRC cohort, serum was collected shortly before hepatic resection of the metastases whereas primary tumor was diagnosed up to six years earlier. Therefore, associations of serum chemerin with tumor stage and grade of CRC were not calculated. In the CRC group, 13 patients received neoadjuvant chemotherapy before liver resection which was, however, not associated with changes in chemerin levels (Figure 1F).

### 2.3. Association of Chemerin with Type 2 Diabetes, Hypertension, Hypercholesterolemia and Hyperuricemia

Circulating chemerin is associated with traits of the metabolic syndrome and some studies described higher levels in type 2 diabetes patients [12,24,30,31]. Chemerin was, however, not increased in those 21 cancer patients with type 2 diabetes compared to patients without this disease (Figure 2A). Chemerin was not changed in the 15 type 2 diabetic HCC patients and the 6 CRC patients when both cohorts were analyzed separately (*p* = 0.25 for HCC and *p* = 0.47 for CRC patients). Likewise, the 9 type 2 diabetes patients of the control group did not have high chemerin serum levels (*p* = 0.09). It should be noted that there were more type 2 diabetic patients in the HCC cohort than in the CRC and control group (Table 1).

Chemerin further regulated blood pressure, and was induced in hypertension [12]. Accordingly, chemerin was higher in the 34 patients with arterial hypertension (Figure 2B). In the HCC subgroup, the 18 hypertensive patients had higher chemerin than the 14 normotensive patients (*p* = 0.02). Moreover, chemerin was elevated in the 16 hypertensive CRC patients when compared to the 20 normotensive patients (*p* = 0.03). Although chemerin positively correlated with systolic blood pressure in the control group (*r* = 0.337, *p* = 0.02) serum levels were not induced in the 18 hypertensive patients (*p* = 0.36).

In addition, hypercholesterolaemic (11 patients) and hyperuricaemic cancer patients (seven patients) had elevated systemic chemerin levels (Figure 2C,D). Again, in the control group chemerin was not changed in the eight patients with hypercholesterolaemia (*p* = 0.65). Distribution of hypercholesterolaemia was comparable in the three groups of patients. Hyperuricemia was only documented in the tumor patients with similar prevalence for CRC and HCC patients (Table 1).

In the HCC group hypercholesterolaemia was diagnosed in three patients and hyperuricaemia in four patients. In the CRC cohort eight patients were hypercholesterolaemic and three were hyperuricaemic. The low number of patients suffering from hypercholesterolaemia and hyperuricaemia in the subgroups may be the reason chemerin changes were not significant (HCC: *p* = 0.13 for hypercholesterolaemia and *p* = 0.12 for hyperuricaemia; CRC: *p* = 0.15 for hypercholesterolaemia and *p* = 0.06 for hyperuricaemia).

The strong association of serum chemerin with comorbidities led us to individually analyze serum chemerin in patients suffering from hypertension, hypercholesterolemia or hyperuricemia and patients, which did not have these comorbidities. In the latter cohort chemerin (*p* = 0.01) and CEA (*p* = 0.02) were higher in CRC patients whereas AFP (*p* = 0.03) was reduced (Figure 3A,B). In the patients suffering from these comorbidities chemerin was similar in both cohorts. CA19-9 (*p* = 0.008) was induced in CRC patients and AST (*p* < 0.001), ALT (*p* = 0.001) and GGT (*p* = 0.003) were lower than in the HCC patients (Figure 3C,D). Above all, chemerin did not differ between patients with liver tumors and non-tumorous controls in both subgroups (Figure 3A,C).

### 2.4. Association of Chemerin with Liver Dysfunction

So far, the association of chemerin with hepatic injury was not resolved [21]. In the control group serum chemerin did not correlate with alanine aminotransferase, aspartate aminotransferase or bilirubin (Table 2). In the tumor patients, serum chemerin was not associated with alanine aminotransferase, aspartate aminotransferase, γ-glutamyltransferase or prothrombin time in the whole cohort, and when CRC and HCC patients were analyzed separately (Table 2). Negative correlations with bilirubin were identified in the whole cohort (Figure 4A and Table 2) and in CRC patients (Table 2).

Chemerin was further not related to alcohol intake which was documented for 24 HCC patients (11 patients did not consume alcohol, five patients rarely drank alcohol, three patients daily had alcohol but less than 15 g and five patients daily had more than 30 g) (Figure 4B).

We additionally evaluated potential associations of serum chemerin with histologic liver abnormalities. In the HCC cohort 16 patients had liver steatosis, 18 had liver inflammation and 23 liver fibrosis. In the CRC group, hepatic steatosis was confirmed by histology in 17 patients, hepatitis in 13 and liver fibrosis in 17 patients. All of these features were comparable in the cohorts. Serum chemerin was, however, not related to any of these traits. Accordingly, serum chemerin did not change with extent of steatosis, inflammation or fibrosis in the whole study group (Figure 5A–C) and when both cohorts were analyzed separately (HCC: *p* = 0.71 for steatosis, *p* = 0.31 for inflammation and *p* = 0.75 for fibrosis; CRC: *p* = 0.77 for steatosis, *p* = 0.44 for inflammation and *p* = 0.05 for fibrosis It is important to note that in the group of tumor patients there was only one patient in the following subgroups: steatosis grade 3, inflammation grade 2 and fibrosis grade 2. Therefore, statistical test is not valid. Chemerin was, however, comparable in patients having no, grade 1 and grade 2 hepatic steatosis. Levels in patients without and grade 1 hepatic inflammation were also comparable. Chemerin in patients without, grade 1 and grade 4 hepatic fibrosis did also not differ. It is thus admissible to conclude that serum chemerin is not associated with hepatic features of liver injury.

## 3. Discussion

This study showed that serum chemerin did not discriminate patients with CRC metastases from HCC patients or controls. Moreover, chemerin levels were not changed with hepatic steatosis, inflammation or fibrosis. TNM stage in HCC patients was not correlated with serum chemerin.

Elevated circulating chemerin in CRC patients was described in recent studies [7,9]. In patients with adenomas serum chemerin was nearly 50% higher than in healthy controls [14]. Chemerin concentration of CRC patients was about 15% increased in one CRC cohort whereas the second analysis reported a more than four-fold induction compared to healthy controls [7,9]. In the present study groups, chemerin serum levels did not differ between CRC patients with liver metastases and patients without tumors. This suggests that chemerin is not solely raised in CRC patients but is high in patients suffering from different diseases. Indeed, higher chemerin was described in psoriasis, inflammatory bowel disease, coronary artery stenosis, obstructive sleep apnea syndrome and chronic obstructive pulmonary disease [32,33,34,35,36]. Therefore, high chemerin may be related to inflammatory processes rather than being a specific marker of CRC. Moreover, chemerin expression was reduced in colon adenomas [13] and may be low in CRC. Therefore, it is unlikely that increased serum chemerin levels do result from the tumor tissue [37]. Serum chemerin comes from adipose tissues and the relationship between fat depots and CRC in tumor patients warrants further investigation.

The mechanisms that contribute to higher serum chemerin in different diseases are presently unknown. Inflammation increased adipocyte chemerin production whereas hepatic synthesis was not changed [21,38,39]. Elevated serum chemerin in obesity did not result in enhanced activation of the chemerin receptor CMKLR1 [40]. Accordingly, C-terminal truncated chemerin isoforms were identified in human obesity [41]. These short variants cannot activate the chemerin receptor [12]. Future work has to examine the factors that influence serum chemerin protein levels and activity in health and disease.

In the tumor patients, serum chemerin was induced in those with hypertension, hypercholesterolemia and hyperuricemia. Accordingly, chemerin was elevated in hypertensive and dyslipidemic patients in different studies [30,42]. Above all, hypertensive or hypercholesterolaemic controls enrolled in the present study did not have higher serum chemerin levels. Notably, a positive correlation of serum chemerin with systolic blood pressure existed in the control cohort. Distribution of these comorbidities was similar between the three cohorts, and thus higher chemerin should have been identified in all groups. This suggests that comorbidity associated induction of chemerin was stronger in tumor patients than controls. Because there were only few patients in some of the subgroups, future studies are needed to validate this suggestion.

Moreover, chemerin did not decline in HCC patients when compared to the non-tumor controls. Chemerin could not discriminate CRC and HCC. HCC patients more often had type 2 diabetes albeit the prevalence of further comorbidities was comparable in both groups. Type 2 diabetes was not linked to higher chemerin in accordance with previous studies [30,42]. Notably, when HCC and CRC patients without hypertension, hyperuricemia or dyslipidemia were compared, chemerin was lower in HCC. In this subgroup besides chemerin, AFP and CEA also differed with the first being higher and the second being lower in HCC. When only patients suffering from these comorbidities were analyzed chemerin was similar in HCC and CRC. Here, transferases were induced in HCC indicating exaggerated liver injury in HCC patients with these complications. Moreover, CA19-9 was higher in the CRC patients.

In the cancer patients, tumor markers CEA and CA19-9 were higher in CRC compared to HCC patients in accordance with previous studies [1,2]. Although tumor markers were not related to comorbidities CEA was only increased in CRC patients without comorbidities, whereas CA19-9 was higher in the CRC patients with secondary complication. Specificities of these markers were thus changed in the two subgroups and future studies have to find out whether this is relevant in the clinical routine. Above all, this analysis showed that chemerin cannot be recommended as a clinical biomarker to discriminate between primary and secondary liver tumors.

A further unresolved issue is whether serum chemerin is a marker of liver injury [21]. In HCC patients chemerin negatively correlated with Child–Pugh score, alanine aminotransferase and bilirubin, and positively with prothrombin time [17]. Associations of chemerin with aminotransferases and bilirubin were not identified in patients with liver cirrhosis [43]. In the cohort studied herein, serum chemerin was negatively correlated with bilirubin in CRC patients whereas associations with further markers of liver health such as aminotransferases and prothrombin time were not identified in any cohort. In addition, chemerin did not change in patients with higher grade of steatosis, inflammation and fibrosis. Limitation of this analysis is that there was only one patient in some of the subgroups and present findings have to be confirmed in the future. Altogether, these preliminary data exclude a strong relation between serum chemerin and liver function.

In line with this suggestion, serum chemerin was not changed in non-alcoholic fatty liver disease (NAFLD) patients with increasing steatosis, inflammation and fibrosis grades [44]. A separate study identified a trend to raised serum chemerin in morbidly obese NAFLD patients with a higher degree of liver steatosis [45]. In a similar patient cohort elevated chemerin was reported in patients with portal inflammation and fibrosis [46]. In chronic hepatitis C serum chemerin was even negatively correlated with necro-inflammatory grade [19].

Serum chemerin was, however, reduced in patients with decompensated liver cirrhosis when compared to patients with compensated disease [43]. Here, a negative correlation with Quick prothrombin time was identified [43]. Coagulation was normal in the patients enrolled in the present study, and all patients had compensated liver cirrhosis. Therefore, a decline of serum chemerin is an indicator of severe liver dysfunction [43], whereas levels are quite normal in patients with compensated disease.

Moreover, chemerin did not correlate with AFP, tumor number or size in a recent study [17]. In the present cohort, chemerin was not associated with AFP, tumor size, grading, TNM stage or vascular invasion. Based on these results it is unlikely that serum chemerin may become a robust biomarker of hepatic injury and HCC.

## 4. Materials and Methods

### 4.1. Patients

Details of the cohorts are summarized in Table 1. Prospective collection of serum of the CRC patients was done from January 2012 to June 2015. Prospective collection of serum of the HCC patients was done from May 2012 to May 2015. Inclusion criteria were histologically confirmed HCC or CRC metastases and age above 18 years. Exclusion criteria was pregnancy.

Liver was histologically examined and scoring was done as suggested by Kleiner et al. [47]. TNM stages were calculated as described [48]. Experiments complied with the guidelines of the charitable state controlled foundation Human Tissue and Cell Research. Each patient signed a written informed consent. The study was approved by the ethical committee of the Regensburg University Hospital (Ethikkommission an der Universität Regensburg) (approval code 15-101-0052, approved on 26 March 2015).

Serum of patients without tumors was obtained from January to June 2008. The cohort included outdoor patients and hospitalized patients who were referred to the interdisciplinary ultrasound department of the University Hospital and was used in previous studies to analyze chemerin and soluble CD163 in serum of controls and patients with non-alcoholic fatty liver disease (NAFLD) [27,28]. Both factors were not changed in the patients with NAFLD [27,28]. The study cohort initially included 56 patients and serum of 49 patients was available for the present study. Patients with hepatobiliary diseases, malignancies, ascites, drugs that cause hepatic steatosis, inflammatory bowel disease, infection with the human immunodeficiency virus, chronic alcohol and drug abuse, familial hyperlipidemia and acute medical conditions with confounding effect on laboratory values, were excluded from the study. All participants signed a form of written consent, and the study was approved by the local Ethics Committee. Aliquots of the sera were stored at −80 °C and freeze-thaw cycles were avoided. It is important to note that serum storage time differed for up to 7 years. This is a limitation of our study. There was no difference in serum chemerin of patients with CRC collected in 2012 (6 patients) and 2015 (10 patients; *p* = 0.1). This suggests that a four year storage period at −80 °C did not grossly affect serum chemerin levels. Chemerin levels may decline during 7 year storage but this effect may be rather small.

### 4.2. Chemerin ELISA

Chemerin ELISA was purchased from R&D Systems (Wiesbaden, Germany) and performed as recommended by the distributor. The plate reader used was the iMark^TM^ Microplate Absorbance Reader (Bio-Rad, Munich, Germany). Absorbance was measured at 450 nm, with the correction wavelength set at 540 nm. Serum was diluted 1:500 fold before analysis.

### 4.3. Laboratory Values

Laboratory values such as bilirubin and tumor markers were routinely measured in the Institute for Clinical Chemistry and Laboratory Medicine, University Hospital Regensburg. Total bilirubin was determined using the Dimension Vista^®^ Flex^®^ reagent cartridge TBIL (Siemens Healthcare Diagnostics Inc., Berkeley, CA, USA). Unconjugated bilirubin was solubilized in a caffeine/benzoate/acetate/ethylene diamine tetraacetic acid mixture. Conjugated bilirubin is soluble in aqueous solvents like water. Solublized bilirubin in serum was coupled with diazotized sulfanilic acid. Thereby diazo-bilirubin was produced. This red chromophore absorbs at 540 nm and was measured by the use of a bichromatic endpoint technique (540 nm, 700 nm).

The ADVIA Centaur^®^ AFP-Assay (Siemens Healthcare GmbH, Erlangen, Germany) and the ADVIA Centaur CEA Assay are sandwich immunoassays with chemiluminescent detection. Two different antigen specific antibodies, a polyclonal rabbit and a monoclonal murine antibody, are used in the assay. ADVIA Centaur^®^ CA 19-9 Assay is a sandwich immunoassay using the same monoclonal antibody in the solid phase and the Lite-reagent.

### 4.4. Statistics

Data are shown as box plots and here the median values, lower and upper quartiles and the range of the values are given. Statistical tests used were Mann–Whitney U Test (to test for significant differences between two independent groups), Spearman correlation (non-parametric correlation analysis), one-way Anova with post-hoc Bonferroni (for comparison of three groups) or Kruskall–Wallis test (for comparison of more than three groups where one of the groups had only 1 patient) (SPSS Statistics 25.0 program, International Business Machines Corporation, Armonk, New York, USA). Chi-square test was used to analyze gender and comorbidity distribution. A value of *p* < 0.05 was regarded as significant. Outliners—greater than 1.5 times the interquartile range—are given as circles, and outliners—greater than 3.0 times the interquartile range—are given as stars.

## 5. Conclusions

Serum chemerin does not discriminate HCC from CRC metastases. Equally important is that levels were neither related to measures of liver injury nor to TNM stage of HCC patients. Chemerin was induced in tumor patients with comorbidities (hypertension, hypercholesterolemia, hyperuricemia), which has to be considered in future clinical studies.

## Figures and Tables

**Figure 1 ijms-20-03919-f001:**
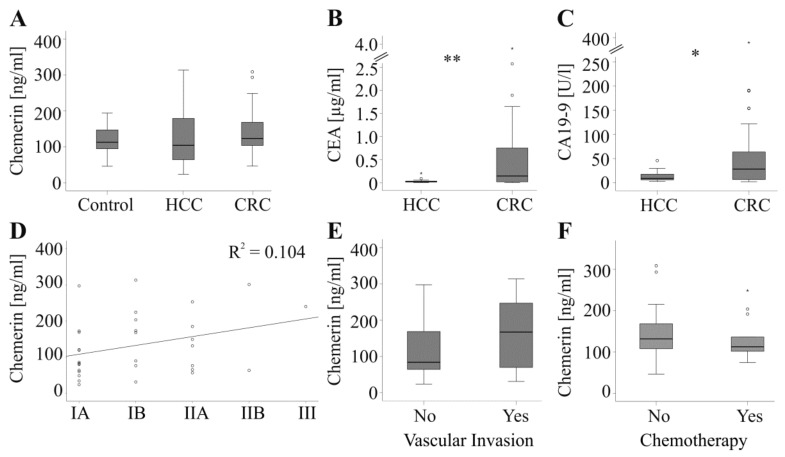
Chemerin and tumor markers. (**A**) Chemerin in serum of 49 controls, 32 patients with hepatocellular carcinoma (HCC) and 36 patients with colorectal carcinoma (CRC). (**B**) CEA in 20 HCC and 32 CRC patients. (**C**) CA19-9 in 19 HCC and 31 CRC patients. (**D**) Correlation of chemerin with TNM stage in HCC patients (TNM stage: IA/IB/IIA/IIB/III, number of patients 14/8/7/2/1). (**E**) Chemerin in 20 HCC patients without and 12 HCC patients with vascular invasion. (**F**) Chemerin in 13 CRC patients with and 23 CRC patients without chemotherapy. * *p* < 0.05, ** *p* < 0.01.

**Figure 2 ijms-20-03919-f002:**
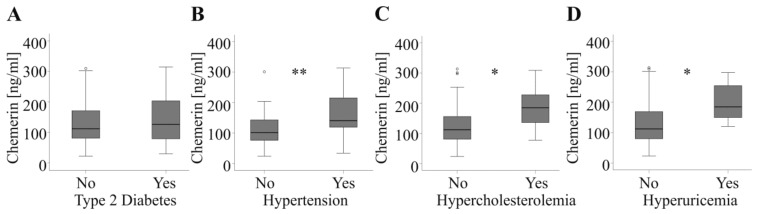
Serum chemerin and comorbidities in tumor patients. (**A**) Chemerin in 21 patients with and 47 patients without type 2 diabetes. (**B**) Chemerin in 34 patients with and 34 patients without hypertension. (**C**) Chemerin in 11 patients with and 57 patients without hypercholesterolemia. (**D**) Chemerin in seven patients with and 61 patients without hyperuricemia. * *p* < 0.05, ** *p* < 0.01.

**Figure 3 ijms-20-03919-f003:**
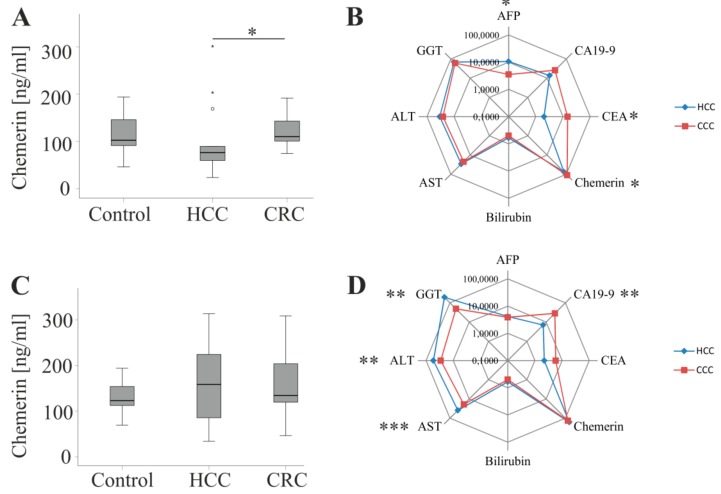
Serum chemerin and comorbidities. (**A**) Chemerin in 27 controls, 14 HCC and 18 CRC patients not suffering from hypertension, hypercholesterolemia or hyperuricemia. (**B**) Spider diagram presentation of chemerin, carcinoembryonic antigen (CEA), cancer antigen 19-9 (CA19-9), alpha-fetoprotein (AFP), γ-glutamyltransferase (GGT), alanine aminotransferase (ALT), aspartate aminotransferase (AST) and bilirubin in the serum of HCC and CRC patients described in A. The spider diagram shows the respective median values on a logarithmic scale. (**C**) Chemerin in 22 controls, 18 HCC and 18 CRC patients suffering from hypertension, hypercholesterolemia or hyperuricemia. (**D**) Spider diagram presentation of chemerin, CEA, CA19-9, AFP, GGT, ALT, AST and bilirubin in the serum of HCC and CRC patients described in C. The spider diagram shows the respective median values on a logarithmic scale. * *p* < 0.05, ** *p* < 0.01, *** *p* < 0.001.

**Figure 4 ijms-20-03919-f004:**
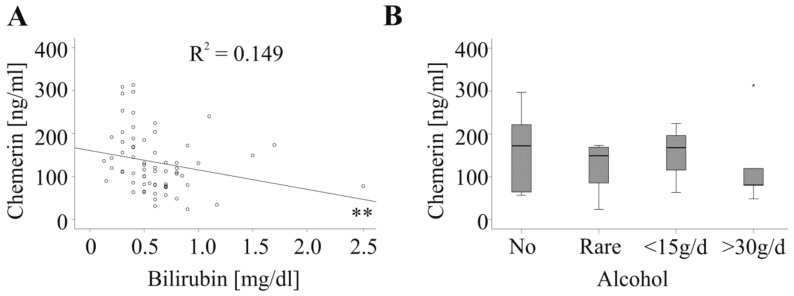
Serum chemerin, bilirubin and alcohol. (**A**) Correlation of chemerin with bilirubin in 31 HCC and 35 CRC patients. (**B**) Chemerin in HCC patients stratified for alcohol intake (No alcohol: 11 patients; Rare: 5 patients; <15 g/d: 3 patients; > 30 g/d 5 patients). ** *p* < 0.01.

**Figure 5 ijms-20-03919-f005:**
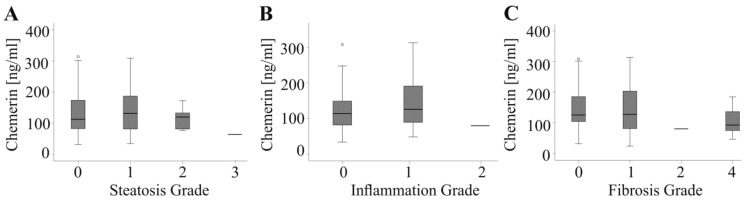
Serum chemerin and liver injury in cancer patients. (**A**) Chemerin in patients stratified for hepatic steatosis (25 patients: no steatosis; 27 patients grade 1; 5 patients grade 2 and 1 patients grade 3; Steatosis grade of 10 patients was not known) (**B**) Chemerin in patients stratified for hepatic inflammation (28 patients: no inflammation; 29 patients grade 1 and 1 patient grade 2. Inflammation grade of 9 patients was not known). *(***C***)* Chemerin in patients stratified for hepatic fibrosis (28 patients: no fibrosis; 25 patients grade 1; 1 patient grade 2 and 14 patients grade 4). Number of patients in some subgroups was 1 and statistical test is not reliable for these subgroups.

**Table 1 ijms-20-03919-t001:** Characteristics of the study group.

Parameter	HCC (32 Patients)	CRC (36 Patients)	Controls (49 Patients)	*p*-Value
Male/Female	27/5	24/12	24/25	#*
Age (years)	63.5 (33.0–85.0)	67.0 (36.0–79.0)^35^	58.0 (21.0–88.0)	
BMI (kg/m^2^)	27.2 (19.7–44.6) ^31^	26.6 (16.3–45.4)	26.2 (20.3–39.7)	
Prothrombin Time (%)	30.8 (26.7–307.0)^30^	28.8 (25.2–39.0)^35^	n.d.	
Bilirubin (mg/dl)	0.6 (0.2–2.5) ^31^	0.5 (0.1–1.0)^35^	0.5 (0.2–1.9)	*
ALT (U/l)	49.5 (17.0–378) ^30^	28.0 (10.0–81.0) ^34^	20.0 (12.0–44.0)	**;#***
AST (U/l)	36.0 (14.0–502.0) ^31^	20.5 (11.0–165.0)^34^	28.0 (20.0–48.0)	*; #*
GGT (U/l)	105 (25–807) ^27^	53 (19–590)^33^	n.d.	**
T2D	15	6	9	**; #**
HC	3	8	8	
HT	18	16	18	
HU	4	3	n.d.	
Tumor Grade: G1/G2/G3	5/20/4^29^	1/23/1^25^		
Primary Tumor T1/T2/T3/T4	13/9/9/1	7/20/3/0^30^		
Vascular Invasion No/yes	20/12	24/2^26^		
TNM Stage IA/IB/IIA/IIB/III/IV	14/8/7/2/1/0	1/4/13/9/1/1/2^31^		

Median values and range, or number of patients per subgroup are shown. Uppercase numbers refer to the patients where this laboratory value / feature was known when data were unavailable for the whole cohort. Reference values for ALT and AST: < 35 U/L for females and < 50 U/L for males, for bilirubin: 0.2–1.4 mg/dL, for GGT: < 40 U/L for females and < 60 U/l for males, for prothrombin time: < 70%. Abbreviations: Alanine aminotransferase, ALT, aspartate aminotransferase, AST; body mass index, BMI; colorectal cancer, CRC; γ-glutamyltransferase, GGT; hepatocellular carcinoma, HCC; hypercholesterolemia, HC; hypertension, HT; hyperuricemia, HU; not documented, n.d.; tumor-node-metastasis, TNM, type 2 diabetes, T2D. The respective *p*-values are listed in the last column of the table. * *p* < 0.05, ** *p* < 0.01 for comparison of CRC and HCC, #* *p* < 0.05, #** *p* < 0.01 and #*** *p* < 0.001 for comparison of controls and HCC patients.

**Table 2 ijms-20-03919-t002:** Correlation of serum chemerin with markers of liver function.

Correlation of Chemerin with:	HCC	CRC	All Tumor Patients	Controls
Prothrombin Time (%)	*r* = −0.103 *p* = 0.587	*r* = −0.262 *p* = 0.128	*r* = −0.172 *p* = 0.170	n.d.
Bilirubin (mg/dL)	*r* = −0.316 *p* = 0.083	***r* = −0.477** ***p* = 0.004**	***r* = −0.386** ***p* = 0.001**	*r* = −0.930 *p* = 0.540
ALT (U/L)	*r* = −0.103 *p* = 0.590	*r* = −0.186 *p* = 0.292	*r* = −0.174 *p* = 0.169	*r* = −0.182 *p* = 0.215
AST (U/L)	*r* = 0.100 *p* = 0.593	*r* = −0.267 *p* = 0.127	*r* = −0.070 *p* = 0.577	*r* = −0.196 *p* = 0.181
GGT (U/L)	*r* = 0.145 *p* = 0.469	r = −0.298 *p* = 0.092	*r* = −0.087 *p* = 0.511	n.d.

Correlation coefficient and *p*-values for the association of chemerin with prothrombin time, bilirubin, alanine aminotransferase (ALT), aspartate aminotransferase (AST) and γ-glutamyltransferase (GGT) are listed for the whole cohort, HCC and CRC patients and controls. Significant correlations are marked in bold. Not defined, n.d.

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
