# Peer review of "Serum Chemerin Does Not Differentiate Colorectal Liver Metastases from Hepatocellular Carcinoma"

_ijms, 2019, doi:10.3390/ijms20163919_

Round 1
Reviewer 1 Report
This is a second version of article entitled: “Serum chemerin does not differentiate colorectal liver metastases from hepatocellular carcinoma”.
All my previous remarks for this article were corrected by authors.
Author Response
All my previous remarks for this article were corrected by authors.
Thank you again for your valuable comments which were very helpful to improve the manuscript.
Reviewer 2 Report
The manuscript has been improved after the first revision. But other minor comments:
The new data introduced in table 1 gives new information. It should be interesting to introduce reference values for Prothrombin time, bilirubin, ALT, AST or GGT.
Line 86-89: the letters are in italics.
Line 105: I would delete the word surprisingly.
Figure 3: I don’t understand the number scale of spider diagram. What is refers to?
Line 317: What wavelength was used to read the Elisa?
Line 303: Did the authors consider that the time of preservation of serum (collected in 2008) could affect the results of the study?
Author Response
We thank the reviewer for the careful reading of our submission and the very helpful and important comments.
The new data introduced in table 1 gives new information. It should be interesting to introduce reference values for Prothrombin time, bilirubin, ALT, AST or GGT.
We included this information in the figure legend
“Reference values for ALT and AST: < 35 U/l for females and < 50 U/l for males, for bilirubin: 0.2 - 1.4 mg/dl, for GGT: < 40 U/l for females and < 60 U/l for males, for prothrombin time: < 70%.“
We also had to correct in the table that prothrombin time is given in % and not s.
Line 86-89: the letters are in italics.
This was corrected.
Line 105: I would delete the word surprisingly.
105 is now line 111, word was removed.
Figure 3: I don’t understand the number scale of spider diagram. What is refers to?
The spider diagram shows the respective median values on a logarithmic scale. This was included in the figure legend.
Line 317: What wavelength was used to read the Elisa?
Please see 4.2.
„Absorbance was measured at 450 nm, with the correction wavelength set at 540.“
Line 303: Did the authors consider that the time of preservation of serum (collected in 2008) could affect the results of the study?
Thank you for this important information. In the literature we found one paper analyzing chemerin before and 18 months after surgery.
“ Thirty-two obese patients undergoing bariatric surgery were tested before and on an average of 18 months after gastric banding or gastric bypass surgery.“ (DOI: 10.1111/j.1365-2362.2010.02255.x).
This group did not report on any differences on chemerin upon storage.
However, we stored serum much longer and there are no data in the literature.
We stored serum at -80°C and avoided freeze-thaw cycles
We now analyzed serum of CRC patients obtained in 2012, 2013, 2014 and 2015. Levels did not differ in 2012 and 2015 and this also indicates that storage time may not grossly affect serum chemerin.
|
Mehrfachvergleiche |
||||||
|
Abhängige Variable: Chemerin |
||||||
|
Bonferroni |
||||||
|
(I) Jahr |
(J) Jahr |
Mittlere Differenz (I-J) |
Std.-Fehler |
Signifikanz |
95%-Konfidenzintervall |
|
|
Untergrenze |
Obergrenze |
|||||
|
12,00 |
13,00 |
-80,26350 |
29,61388 |
,064 |
-163,5479 |
3,0209 |
|
14,00 |
-22,15529 |
25,02829 |
1,000 |
-92,5434 |
48,2329 |
|
|
15,00 |
16,62650 |
26,48746 |
1,000 |
-57,8653 |
91,1183 |
|
|
13,00 |
12,00 |
80,26350 |
29,61388 |
,064 |
-3,0209 |
163,5479 |
|
14,00 |
58,10821 |
25,02829 |
,161 |
-12,2799 |
128,4964 |
|
|
15,00 |
96,89000* |
26,48746 |
,005 |
22,3982 |
171,3818 |
|
|
14,00 |
12,00 |
22,15529 |
25,02829 |
1,000 |
-48,2329 |
92,5434 |
|
13,00 |
-58,10821 |
25,02829 |
,161 |
-128,4964 |
12,2799 |
|
|
15,00 |
38,78179 |
21,23721 |
,463 |
-20,9445 |
98,5081 |
|
|
15,00 |
12,00 |
-16,62650 |
26,48746 |
1,000 |
-91,1183 |
57,8653 |
|
13,00 |
-96,89000* |
26,48746 |
,005 |
-171,3818 |
-22,3982 |
|
|
14,00 |
-38,78179 |
21,23721 |
,463 |
-98,5081 |
20,9445 |
|
|
*. Die Differenz der Mittelwerte ist auf dem Niveau 0.05 signifikant. |
||||||
We added this paragraph in 4.1.
“Aliquots of the sera were stored at -80 °C and freeze-thaw cycles were avoided. It is important to note that serum storage time differed for up to 7 years. This is a limitation of our study. There was no difference in serum chemerin of patients with CRC collected in 2012 (6 patients) and 2015 (10 patients; p = 0.1). This suggests that a four year storage period at -80°C did not grossly affect serum chemerin levels. Chemerin levels may decline during 7 year storage but this effect may be rather small. “